# Virus Behavior after UV_254_ Treatment of Materials with Different Surface Properties

**DOI:** 10.3390/microorganisms11092157

**Published:** 2023-08-25

**Authors:** Castine Bernardy, James Malley

**Affiliations:** Department of Civil and Environmental Engineering, College of Engineering & Physical Sciences, University of New Hampshire, Durham, NH 03824, USA; castine.bernardy@unh.edu

**Keywords:** UV_254_, disinfection, inactivation, ultraviolet, dose-response, surface, MS-2 bacteriophage

## Abstract

The COVID-19 pandemic highlighted the limitations in scientific and engineering understanding of applying germicidal UV to surfaces. This study combines surface characterization, viral retention, and the related UV dose response to evaluate the effectiveness of UV_254_ as a viral inactivation technology on five surfaces: aluminum, ceramic, Formica laminate, PTFE and stainless steel. Images of each surface were determined using SEM (Scanning Electron Microscopy), which produced a detailed characterization of the surfaces at a nanometer scale. From the SEM images, the surface porosity of each material was calculated. Through further analysis, it was determined that surface porosity, surface roughness, contact angle, and zeta potential correlate to viral retention on the material. The imaging revealed that the aluminum surface, after repeated treatment, is highly oxidized, increasing surface area and surface porosity. These interactions are important as they prevent the recovery of MS-2 without exposure to UV_254_. The dose response curve for PTFE was steeper than ceramic, Formica laminate and stainless steel, as inactivation to the detection limit was achieved at 25 mJ/cm^2^. These findings are consistent with well-established literature indicating UV reflectivity of PTFE is maximized. Statistical testing reinforced that the efficacy of UV_254_ for surface inactivation varies by surface type.

## 1. Introduction

The US CDC defines disinfection as the process that eliminates many or all pathogenic microorganisms, except bacterial spores, on inanimate objects [1]. Ultraviolet (UV) wavelengths are an effective disinfectant for microorganisms. UV damages the DNA/RNA of bacteria, fungi, and viruses. This damage is referred to as inactivation and renders the microbe unable to reproduce, most commonly through the creation of thymine or cytosine dimers for DNA or uracil dimers for RNA. UV technologies have been utilized for water disinfection purposes since 1910, although the UV surface disinfection field is still lacking in original, specific research findings and understanding [2].

A June 2023 workshop sponsored by International Ultraviolet Association (IUVA), the American Society of Heating, Refrigerating, and Air-Conditioning Engineers (ASHRAE), the American National Standards Institute (ANSI) and the National Institute of Standards and Technology (NIST) identified the emerging importance and knowledge gaps of increasing the application of germicidal UV to public spaces to reduce the morbidity and mortality caused by viral epidemics and pandemics [3]. Prior to the COVID-19 pandemic, UV surface disinfection received little attention. The pressure to manufacture surface disinfection products is expected to increase the UV market from USD 4.8 billion to a USD 9.2 billion industry in 2026 [4]. As a result, the utilization of UV disinfection technologies for surface treatments has increased in the United States and around the globe. Even as COVID-19 cases decline, the mortality resulting from annual outbreaks of flu highlights the need to improve our scientific and engineering knowledge on the effects of surface characteristics on UV disinfection efficacy.

Without a proper validation protocol, the products created and marketed for surface disinfection may be ineffective or even a threat to human health. A whitepaper compiled by the IUVA highlighted the state of the research and provided research suggestions to assist in closing knowledge gaps. The whitepaper suggested that understanding the effects of surface roughness, hydrophobicity, reflectivity, photochemical interactions, irradiance, and dose distribution is prudent for developing safe and effective surface disinfection products [5].

UV disinfection can be successfully utilized as another barrier to limit the spread of infectious diseases and healthcare-associated infections (HAIs). The healthcare industry is interested in upper air units, mobile units, HVAC, and fixed/timed disinfection units [6]. Poster et al. [7] report that prior to full adoption of these technologies in the healthcare industry, more research in UV theory, safety, reliability, performance, and simulations must be conducted. In addition, before patients and healthcare workers feel comfortable using these devices more information must be available including data regarding how to effectively monitor the UV dose delivered and how to select the most appropriate device [7]. The healthcare industry has called for the development of a standardized protocol to address these concerns before they can be confident in the safety and effectiveness of UV-C surface disinfection devices [7,8,9].

A 2019 paper evaluated UV as a disinfection technology to reduce HAIs [8]. The study emphasized that the factors to be considered for surface disinfection efficacy vary from those affecting water disinfection. These factors include surface distance from the UV lamp, surface contact angle of the UV ray and line of sight (shadowing). UV disinfection efficacy decreases when the line of sight is affected. This could be due to the surface topography, shape, porosity, or lamp orientation relative to the surface. A method to increase the line of sight (decrease shadowed areas) is by implementing reflective surfaces into the design [8].

The effects of shadowing were observed in another study utilizing the “Nanoclave Cabinet” UV device to inactivate *Methicillin-resistant Staphylococcus aureus* (MRSA), *Vancomycin Resistant Enterococcus* (VRE), *Acinetobacter baumannii* and *Klebsiella pneumoniae* (10^6^ cfu/cm^2^ inoculum) [10]. The surfaces evaluated included a blood pressure gauge, patient call button, infusion pump, tympanic thermometer, oximeter, computer keyboard/mouse, TV remote control, and blood pressure cuff, and were treated with a UV dose of 318 mJ/cm^2^. The blood pressure cuff and tympanic thermometer yielded the lowest inactivation levels with all four bacterial species evaluated. This was thought to be a result of the shape of the objects prohibiting the UV rays from contacting the full surface. Regardless of the attempts to incorporate reflective materials into the design, shadowing adversely affected UV disinfection efficacy. Further analysis into the mechanistic effects of the characteristics of each surface on UV_254_ disinfection efficacy was not conducted [10].

Finally, the effects of surface type (specifically porous versus non-porous) on UV-C inactivation of SARS-CoV-2 have been evaluated [11]. The study looked at thirteen surface types and found that the UV_254_ dose required for 2–3 log inactivation of SARS-CoV-2 varied significantly by surface type. Most notably, a 176-fold higher UV_254_ dose was required to achieve 2 log inactivation of car upholstery surfaces (porous) when compared to polystyrene (non-porous). In addition, it was discovered that surfaces with high cotton percentages require significantly higher UV doses to achieve the same level of inactivation of surfaces with lower percentages of cotton contents. This is thought to be caused by the retention of SARS-CoV-2 particles within the cotton fibers. The inactivation level observed on the leather surface was surprising, as it behaved similarly to the non-porous surfaces. This study emphasizes that the effects of surface type on inactivation efficacy are “highly variable and composition-dependent” [11]. Therefore, these results call for more thorough research to uncover the causal mechanisms behind this variation. Additionally, Kowalski [2] reports that the UV penetration depth is minimal for most materials, for polymers, the depth is approximately 25 µm [2]. As cotton is a natural polymer [12], this suggests that the inactivation of the cotton took place only on the immediate surface.

The research discussed in this paper assists in closing the current knowledge gaps in the industry by providing novel findings regarding the effects of surface type on UV_254_ inactivation efficacy. This research addresses the effectiveness of UV_254_ inactivation of MS-2 bacteriophage on aluminum, ceramic, Formica laminate, PTFE and stainless steel, as these surfaces have varying properties and practical uses. The objective of this study was to conduct an in-depth analysis of surface characteristics that have not been thoroughly studied, such as surface porosity, surface roughness, hydrophilicity (contact angle), zeta potential, and surface charge and their resulting effects on UV_254_ inactivation of MS-2 bacteriophage. This paper uses an engineering approach to quantify these surface characteristics to gain an understanding of the mechanisms dominating UV inactivation efficacy by surface type. Aluminum, ceramic and PTFE were selected as they represent a range in these characteristics, whereas Formica laminate and stainless steel were also added to the project to represent other materials that are also commonly used in healthcare facilities. 

This work provides the UV industry with critical information regarding the effects of surface type on UV_254_ efficacy. These results assist in explaining the interactions of surface, virus, and UV effectiveness, which point to the importance of surface characterization when developing a protocol for the validation of surface inactivation applications. The findings provide insight into which surfaces are best suited for UV inactivation. This research provides major contributions to the development of validation protocols for the UV surface inactivation industry which will lead to greater protection of public health.

## 2. Materials and Methods

This research utilized a controlled environmental chamber to conduct experiments. The chamber temperature was 28 degrees Celsius with a relative humidity of 55%, resulting in a dew point of 18 degrees Celsius. These experiments used a 254 nm collimated beam device made by Trojan Technologies to evaluate the efficacy of UV inactivation on five surface types. The surfaces evaluated include aluminum, ceramic, Formica laminate, PTFE, and stainless steel. Table 1 below displays detailed information regarding the manufacturer and composition of the surfaces utilized in this research.

Prior to experimentation, the average irradiance received by the surfaces was determined. A 10 cm by 10 cm grid was created and irradiance values were recorded with a calibrated NIST traceable IL1700 radiometer every 2 cm in the X and Y direction. The measurement of UV dose conformed with and employed NIST 20/O05 UV Radiometric-Standard Detector and NIST 20/O06 UV Radiometric-Standard Sources. Figure 1 below displays the spread of irradiance emitted by the Trojan collimated beam unit and was used to determine a weighted average of the irradiance received by the surfaces. 

The distance of the collimated beam above the radiometer (9 cm) was selected as it provided the most even distribution of irradiance. The weighted average irradiance value was calculated at 0.203 mW/cm^2^ and was used to determine the required exposure times (Table 2) for each dose. Each surface was exposed to five UV doses, ranging from 0–100 mJ/cm^2^. The equation used to calculate UV dose is shown below.
UV Dose (mJ/cm^2^) = UV Irradiance (mW/cm^2^) × Exposure Time (seconds).(1)

The UV doses selected were based on the vastly reported MS-2 inactivation data in water samples tabulated by Malayeri et al. [13]. The MS-2 stock solution had a concentration of approximately 10^9^ PFU/mL and was transferred to the surfaces with a cotton swab. The swab was submerged in the stock solution and evenly applied to each surface, using a 5 cm diameter stencil. Several methods to apply the virus to the surfaces were thoroughly evaluated, including using a spray bottle and nebulizer. The most reproducible results were achieved using the cotton swab inoculation method.

The surfaces were placed under the UV-collimated beam, where they received their respective dose. The surfaces were not allowed to dry prior to starting the experiments. The effects of inoculum dry time before UV treatment have been reported in the literature [14,15,16]. SARS-CoV-2 studies have been conducted and suggest that the inactivation rate of SARS-CoV-2 is unaffected by drying prior to UV treatment [14,15]. This was likely due to the short dry times and UV dosing times utilized for these experiments. A study using pulsed UV for inactivation of SARS-CoV-2 and MS-2 Bacteriophage found varying results. The SARS-CoV-2 achieved higher inactivation with the wet inoculum, yet no effect of dry time was observed for the MS-2 results [16]. This work may have yielded different results if the surfaces were allowed to dry prior to UV treatment.

After receiving their respective UV dose, the surfaces were rinsed with 50 mL of sterile phosphate buffer solution (PBS) using a pointed spray nozzle. A clamp was used to hold the surface above a beaker, which collected the PBS rinse water. The positive control (0 mJ/cm^2^) experiments were immediately rinsed with 50 mL of PBS without exposure to the UV-collimated beam. Three trials were conducted for each surface, where each UV dose was evaluated as a separate experiment.

The PBS rinse water from each experiment was sent to a partnering lab (GAPLAB, London, ON, Canada) for analysis using plaque assay to determine MS-2 bacteriophage infectivity. For QA/QC purposes each trial was split into duplicates for plaque assay analysis; therefore, each UV dose has a total of six data points. 

The surfaces and glassware utilized for the experiments were sterilized to prevent contamination. The metal surfaces (aluminum and stainless steel), glassware and phosphate buffer solution were autoclaved prior to experimentation. The remaining surfaces (ceramic, Formica laminate, and PTFE), plastic stencils, and clamps were sterilized in a bleach solution. After their 20-min contact time, they were thoroughly rinsed with tap water, followed by RO water. These items were then set out to dry before the next experiment. Negative controls were conducted on the autoclaved and bleached materials, revealing MS-2 bacteriophage concentrations below the detection limit. These data can be located in the compiled data DOI listed at the end of the manuscript.

### 2.1. MS-2 Bacteriophage

The surrogate used for experimentation was MS-2 bacteriophage. MS-2 was chosen for its reliability and lab-to-lab reproducibility. This virus is not pathogenic, therefore can be utilized safely in the University’s BSL-2 laboratory. 

Additionally, MS-2 bacteriophage is believed to be an adequate surrogate for SARS-CoV-1 and SARS-CoV-2 [17]. Utilizing MS-2 for experimentation was advisable as it is the most common surrogate used in the validation of UV systems. 

The MS-2 samples analyzed via plaque assay were sent to GAPLAB in Ontario, Canada. The MS-2 strain used was ATCC 15597-B1 and the plating host was *E. coli F Amp* ATCC 700891. The MS-2 was enumerated (plaque assay) using a single agar layer method modified from Standard Methods 9224E [18].

The MS-2 stock was propagated by spiking 1 mL of MS-2 daughter stock (1 generation removed from ATCC stock) into a shaking *E. coli Hfr* ATCC 15597 in tryptic soy broth culture (~4.5 h). The combined stock was allowed to continue shaking overnight at 35 °C. It was then centrifuged to remove cellular debris. An aliquot of the MS-2 was taken, and a seven point UV dose-response curve from 0–100 mJ/cm^2^ was generated using a collimated beam to ensure it falls within the internal laboratory QC limits [19,20].

### 2.2. Aluminum Agitation Experiments

For these experiments, one aluminum disk was inoculated with MS-2 bacteriophage via the swabbing method described previously. The disk was rinsed with a 50 mL aliquot of sterile PBS and the surface was agitated with a clean cotton swab to dislodge the virus. The cotton swab was then submerged in the PBS wash water. This process was repeated five times in total; therefore, the disk was rinsed with a total of 250 mL of PBS and agitated with five cotton swabs. All the aliquots were analyzed using plaque assay methods.

### 2.3. Rate Constants (k Values)

The UV_254_ dose response curves are fit with polynomial trendlines, which are displayed in the figures. The rate of change was determined for each interval and then averaged to find the rate of change of the polynomial trendline (k value). For each x value (UV dose), the y value was calculated using the figures’ corresponding polynomial equation. The change in y was divided by the change in x to find “k” at each interval. These values were averaged to find the rate of change of the polynomial.

The PTFE data displays a linear fit, therefore the k value was determined in the equation for the trendline. The maximum inactivation rate is a measure of inactivation without the effects of tailing. Tailing data was excluded from these rates by removing data that had overlapping error bars.

### 2.4. Contact Angle

The contact angle measurements were taken with the Biolin Scientific ThetaLite 101 tensiometer. The OneAttension software (Version 3.2 (r5971)) was utilized for collecting data from the optical tensiometer. Depending on the variability of contact angle values, 5–10 data points were collected. The average droplet volume applied to the surfaces for contact angle measurements was 10.6 µL.

### 2.5. Surface Roughness

The surface roughness measurements were taken with an Olympus LEXT OLS5000 SD Laser Confocal Microscope. The data points collected from the confocal microscope measured the average height variation from the mean center line of the surface. These values were calculated across the face of the surface.

### 2.6. SEM and Surface Porosity

A Tescan Lyra3 GMU FIB Scanning Electron Microscope (SEM) was utilized to generate magnified images of the five surfaces, such that pore sizes could be quantified. The SEM machine measured the surfaces at 100,000 times magnification.

The surface porosity for each material was determined using NIS-Elements AR 5.14.01 software to quantify the area of depressions (pores) from each SEM image. The porous area was divided by the total surface area to find the surface porosity of each material.

### 2.7. Zeta Potential

Prior to analysis, the surfaces were machined to create three 15 mm diameter samples for zeta potential analysis. The samples were then soaked in a phosphate buffer solution (pH 7.2) for sixty minutes. The samples were then air-dried. After drying, they were rinsed with type I lab water to remove salt deposits. 

The samples were then analyzed with an Anton Parr Model SurPASSTM3-Standard fitted with the Adjustable Gap Cell. The SurPASSTM 3 utilizes electrokinetic analysis that combines the classic streaming potential and streaming current methods for direct analysis of the zeta potential of macroscopic surface samples under controlled pH and ionic strength conditions. The SurPASSTM 3 was calibrated per manufacturer specifications using NIST traceable standards over a range of +100 to −100 mV. SurPASS3 pH monitoring ensured that the pH of the analysis was maintained at 7.2 ± 0.1.

### 2.8. Statistical Analysis

Statistical analyses were conducted via equivalency testing in the statistical software JMP Pro 16. The k values (slope) of the dose-response curves were analyzed to determine the statistical significance of UV_254_ inactivation efficacy by surface type. Equivalence testing is an extension of hypothesis testing, but is preferred for scientific applications, as it incorporates an additional layer to evaluate if the results are of scientific relevance. In the JMP Pro 16 software, the user can enter a value that is “the practical limit of significance” for their data set. This approach evaluates statistical significance and relevant significance to the surface inactivation field [21].

The equivalence tests were run three times using different “practically zero” values, which were determined by data sets published in the literature [2,13,22]. Practical equivalence was determined by *p*-values below 0.05. These values were taken from peer-reviewed publications and standards as an acceptable range of k values for MS-2 inactivation in water. The difference between the upper and lower standard deviations of the California NWRI guidelines [22] and the collected MS-2 data of Malayeri et al. [13] were 0.00065 and 0.0032, respectively. An additional “practically zero” k value was determined from the standard deviation of k values listed in the Ultraviolet Germicidal Irradiation Handbook; this value was 0.0102 [2]. See the Appendix A section for further details and JMP outputs.

## 3. Results

### 3.1. Aluminum

The inactivation efficacy of UV_254_ for the virus was evaluated for five surface types. The data below (Figure 2) displays the results of MS-2 inactivation on the aluminum surface for UV_254_ doses 0–100 mJ/cm^2^. 

The results of these experiments (Figure 2) indicate that the log loss of MS-2 bacteriophage is consistently high for all doses evaluated. The MS-2 bacteriophage loss was at or just below the method detection limit for all UV doses evaluated, 0–100 mJ/cm^2^. The error bars are not visible on this graph due to the low variability observed between each trial.

Figure 2 displays a 6.2 log reduction in MS-2 bacteriophage plaques at the positive control data point (0 mJ/cm^2^). This finding suggests that the dominant viral loss/inactivation mechanism is an interaction involving the aluminum surface, rather than the UV doses applied. 

To further evaluate the theory that viable MS-2 bacteriophage cannot be recovered from this aluminum surface, a swabbing and rinsing experiment was designed. The aluminum surface was agitated repeatedly to recover additional infective MS-2 bacteriophage. These data are shown in the Appendix A. The experiment was conducted in an analogous manner to the positive control experiments, except that serial washing and agitation with a cotton swab took place to desorb the virus. Like Figure 2, these experiments failed to display any recovery of infective MS-2 and revealed minimal variability between trials.

This experiment provides further evidence that a significant interaction occurs between the MS-2 bacteriophage virus and the aluminum surface. The high repeatability of these data points provides confidence that an interaction is occurring between the aluminum surface and MS-2 bacteriophage, warranting further exploration.

### 3.2. Ceramic 

The data below (Figure 3) displays the results of MS-2 inactivation on the ceramic surface for UV_254_ doses 0–100 mJ/cm^2^. 

The results of these experiments are not equivalent (see Appendix A for information on equivalency testing) to the aluminum results and follow a more typical dose-response curve pattern commonly seen with microbial inactivation work. The data displays a strong correlation (R^2^ = 0.998) between UV dose and log MS-2 bacteriophage loss. As the UV dose increased, log MS-2 loss increased. This trend continues until it reaches a plateau at a UV dose of 75 mJ/cm^2^.

These data were adjusted to account for viral retention on the ceramic surfaces. A sharp increase in log inactivation is apparent from 0–50 mJ/cm^2^. The rate of inactivation between 50 and 75 mJ/cm^2^ decreases to 0.61 log. No change in inactivation is observed between 75 and 100 mJ/cm^2^, indicating an inactivation plateau. The maximum MS-2 bacteriophage inactivation was 4.4 log and was achieved at 75 and 100 mJ/cm^2^.

### 3.3. Formica Laminate 

The data below (Figure 4) displays the results of MS-2 inactivation on the Formica laminate surface for UV_254_ doses 0–100 mJ/cm^2^.

The data shown in Figure 4 displays a strong correlation (R^2^ = 0.995) between UV dose and log MS-2 bacteriophage loss. As the UV dose increased, the loss of MS-2 bacteriophage increased. The maximum inactivation observed was 4.7 log, which occurred after the 100 mJ/cm^2^ UV dose. Three trials were conducted, and little variability was observed between trials, as shown by the error bars. These data were adjusted to account for viral retention on the Formica laminate surfaces. 

### 3.4. PTFE

The data below (Figure 5) displays the results of MS-2 inactivation on the PTFE surface for UV_254_ doses 0–100 mJ/cm^2^.

The PTFE surface data (Figure 5) displayed a significantly different UV dose-response curve than the other surfaces experimented on (Appendix A). UV_254_ doses 0–100 mJ/cm^2^ were evaluated. Due to the sharp increase in inactivation on this surface, additional trials were evaluated at 10 mJ/cm^2^. 5.6 log MS-2 inactivation, the detection limit for these experiments, was achieved at 25 mJ/cm^2^. 

Like the other surfaces, the data was adjusted to account for viral retention on the PTFE surface. The PTFE surface yielded MS-2 inactivation to the detection limit at a UV dose of 25 mJ/cm^2^. To create a UV dose-response curve, additional experiments were conducted at 10 mJ/cm^2^. The UV dose-response curve from 0–25 mJ/cm^2^ was linear, with a strong correlation (R^2^ = 0.999), such that as the UV dose increased, log MS-2 bacteriophage dramatically increased. MS-2 bacteriophage could not be recovered from the PTFE surface after receiving a 25 mJ/cm^2^ UV dose.

Unlike the results of the aluminum surface, MS-2 loss was not observed at the positive control (0 mJ/cm^2^) data points. This suggests that the high inactivation levels observed are a result of the UV_254_ doses received, rather than an interaction between surface and virus.

The PTFE achieved the highest inactivation level of all surfaces experimented on. These results indicate that the PTFE surface interacts with UV_254_ wavelengths, resulting in significantly higher levels of inactivation, warranting further discussion.

### 3.5. Stainless Steel

The data below (Figure 6) displays the results of MS-2 inactivation on the stainless steel surface for UV_254_ doses 0–100 mJ/cm^2^. 

The data shown in Figure 6 displays a strong correlation (R^2^ = 0.991) between UV dose and log MS-2 bacteriophage loss. As the UV dose increased, the loss of MS-2 bacteriophage increased. The maximum level of inactivation on the surface was 4.6 log, which occurred after the 100 mJ/cm^2^ UV dose. 

This data was adjusted to account for viral retention on the stainless steel surfaces. These data appear like the Formica laminate results (Figure 4), such that as UV dose increases, log inactivation of MS-2 increases. Although, these data have higher variability than the Formica laminate results. 

The dose-response curves displayed in Figure 2, Figure 3, Figure 4, Figure 5 and Figure 6 highlight the importance of surface type on UV_254_ inactivation efficacy. Further surface characterization was performed to gain a better understanding of the mechanisms responsible for altering the dose-response curves by surface type. These values were tabulated and are shown in Table 3 below. 

Equivalency testing (Appendix A) was conducted on the inactivation rate constants (k values). Three trials of equivalency testing were conducted. The input values required to test for statistical equivalency were tabulated from data sets published in the literature [2,13,22], and the results of these tests can be found in Appendix A. Two of the equivalency tests revealed no practical significance and one test revealed practical significance for the inactivation constants of Formica laminate and stainless steel.

The surface characteristics reported in Table 3 are shown below. These figures display MS-2 bacteriophage recovery as a function of contact angle, surface porosity, surface roughness and zeta potential.

### 3.6. Contact Angle

Data were collected on the contact angle of each surface and are displayed graphically as a function of MS-2 recovery (Figure 7). 

The contact angle for aluminum, ceramic, Formica laminate, PTFE, and stainless steel and their corresponding MS-2 recovery is reported in Figure 7. The recovery of MS-2 (PFU/mL) was tabulated from the positive control values (0 mJ/cm^2^) for each surface. The aluminum and PTFE surfaces displayed the lowest and highest contact angle and MS-2 recoveries, respectively. Figure 7 displays a strong correlation (R^2^ = 0.955) between MS-2 bacteriophage recovery and contact angle. These data suggest that as the contact angle increases, the recovery of MS-2 bacteriophage increases. The figure suggests that surfaces with high contact angles retain fewer viral particles, therefore allowing more MS-2 to be extracted from the surface. 

### 3.7. Surface Roughness

Further studies (Figure 8) were conducted to determine if surface roughness affected surface viral recovery.

Figure 8 displays viral recovery as a function of surface roughness. The recovery of MS-2 (PFU/mL) was tabulated from the positive control data (0 mJ/cm^2^) from each surface. The data displays a strong correlation (R^2^ = 0.999) between surface roughness and recovered MS-2 bacteriophage for PTFE, Formica laminate, and ceramic. These surfaces display an inverse relationship, such that as surface roughness increases, MS-2 bacteriophage recovery decreases. Stainless steel disks were excluded from this figure since the stainless steel surface was highly polished therefore the surface roughness measurements were at the detection limits of the test. As previously discussed, MS-2 bacteriophage could not be recovered from the aluminum surface; therefore, these data were also excluded. 

### 3.8. SEM

Data was collected using Scanning Electron Microscopy (SEM) to view the pores of the surfaces. The surfaces at 100,000 times magnification are shown in Figure 9.

The magnified surfaces displayed in Figure 9 outline the depressions (green) on each surface. The depressions indicate pore spaces and vary by surface type. The aluminum surface had the greatest porous area, followed by ceramic. Aluminum surfaces are a well-studied material since both matte and anodized forms are commonly used in the healthcare industry and in HVAC systems [23]. The ceramic surface had many crevices between pieces of material, which contributed to most of its porous area. Formica laminate, PTFE, and stainless steel had very few porous areas. 

From these data, the percentage surface porosity was determined for each material. Aluminum and ceramic had the highest surface porosity, whereas the surface porosity of Formica laminate, PTFE, and stainless steel was small. This can be observed in Figure 9 by the larger area of highlighted depressions for aluminum and ceramic. The relationship between viral recovery and surface porosity is shown below in Figure 10.

Figure 10 displays a strong correlation (R^2^ = 0.858) between the recovery of MS-2 bacteriophage (PFU/mL) and surface porosity. Data from the positive control experiments (0 mJ/cm^2^) was tabulated to compare MS-2 concentration recovery versus surface porosity. The figure displays an inverse relationship, such that as surface porosity decreases, viral recovery increases.

Figure 11 displays a strong correlation (R^2^ = 0.903) between the recovery of MS-2 bacteriophage (PFU/mL) and zeta potential. Data from the positive control experiments (0 mJ/cm^2^) was tabulated to compare MS-2 concentration recovery versus zeta potential. The figure displays an inverse relationship, such that as zeta potential increases, viral recovery decreases. 

## 4. Discussion

The findings of this research highlight that the efficacy of UV_254_ as an inactivation technique varies significantly by surface type. Information regarding surface type and characteristics must be included in the development of validation protocols for UV surface inactivation technologies. Understanding the key mechanisms responsible for viral retention, recovery, and inactivation is critical to developing better strategies for UV_254_ inactivation of surfaces. 

The k values were calculated from the polynomial trendlines displayed in Figure 3, Figure 4 and Figure 6. Polynomial trendlines help to illustrate the different regions of the UV dose-response curves (shoulder, exponential and tailing) [19]. However, the maximum inactivation rates also provide valuable information about the behavior of the virus, as they exclude the effects of tailing. The k values from both types of trendlines are shown in Table 3. The maximum inactivation rates are higher than the k values reported for the full dose-response curves, as they only represent the inactivation of the accessible virus. Mattle et al. [24] studied the effects of tailing during UV_254_ inactivation of MS-2 bacteriophage. The authors found that after exponential decay, the inactivation rate decreases due to clumping (aggregation) and recombination. Our data reiterates these findings. The surface characteristics, such as porosity likely increased the shielding of the virus. This can be observed by the significant tailing shown in Figure 3. The effects of surface characteristics are discussed in detail below.

The aluminum results (Figure 2) were most surprising, as the UV industry has a long history of using aluminum materials. It was determined by the positive control data that viral surface interactions are the causal mechanism for MS-2 bacteriophage removal/inactivation, which can be explained by data from the literature [25,26,27]. The highly oxidized aluminum surface which resulted from repeated autoclaving yielded a very porous and reactive material for virus sorption. Interactions between viral particles and charged surfaces are dominated by their electrostatic attraction. Due to the strong positive charge of aluminum [25], it is likely that an electrostatic force between the virus and the surface creates a strong adsorptive bond, thus affecting viral recovery [26].

Many studies suggest that aluminum interactions inactivate viruses during water treatment processes. For example, a 1988 study reported on the inactivation of poliovirus due to aluminum exposure within a water column [27]. The study stated that 99.9% of the virus (titer of 2.2 × 10^7^ PFU/mL) was adsorbed to the aluminum within a 2-h contact time. After 76 h, 93% of the poliovirus had been desorbed, although was non-infective. It is thought that the initial attraction of the virus to the aluminum is due to the negative charge of the virus and the positive charge of the aluminum. As the contact time increases, it is thought that the oxide coating on the aluminum causes damage to the peptide backbone. Over time the peptide fragments/ and or cleaved bonds can be reabsorbed into the solution, therefore allowing for the detection of inactivated poliovirus [27].

Another similar theory discussed in this paper addressed the possibility that the strength of the electrostatic forces between the virus and aluminum is strong enough to dissociate the capsid. The author suggested that the electrostatic forces are stronger than the forces holding the viral capsid together [27]. Both methods result in a significant loss in infectivity. In addition, it should be noted that these mechanisms are predicted to lose their effectiveness over time, due to the remaining partial sorption of viral particles [27].

Matsushita et al. [25] examined the viricidal effect of aluminum coagulants on four viruses (Qβ, MS-2, T4, and P1). The study used four aluminum coagulants and then dissolved the floc into an alkaline solution. Although the results differed by virus and coagulant type, the study revealed that all aluminum-based coagulants caused viral inactivation to varying degrees for all viruses evaluated. The proposed mechanism responsible for the virucidal activity of the aluminum coagulants is irreversible adsorption. It is thought that the aluminum species bonds to viral lipoproteins and binding structures. This binding action results in the inactivation of the virus [25].

The results from the studies conducted by Thurman et al. [27] and Matsushita et al. [25] provide insight into the mechanisms behind the results seen in Figure 2. As discussed, aluminum has a positive charge, and the charge of MS-2 bacteriophage is negative. Therefore, the aluminum and MS-2 particle bond ionically, not releasing the MS-2 back into the PBS solution when rinsed. In addition, an oxide coating is created on the aluminum surface after exposure to water. If the virus can be desorbed from the surface, the outer proteins of the virus are damaged due to the strong ionic bond between the virus and aluminum or the oxidizing species on the surface. The damage to these proteins prohibits the virus from infecting the host (*E. coli*), therefore producing non-detectable infectivity results. 

Although these results discuss the probable mechanisms controlling MS-2 bacteriophage inactivation on aluminum, they may be transferrable to viruses without a protein coat as well. Viruses without protein coats, such as viroids [28] and Hepatitis D [29] may yield similar results. The key protein damage credited for the viral inactivation mentioned in the studies above is not possible for these viruses. Although, both viruses are RNA viruses, which have a highly negative charge [30]. The positive charge of the aluminum and negative charge of the RNA of these viruses may result in ionic bonding similar to the observed bonds that occur between the protein-coated viruses and the aluminum. It is likely that these viruses would be retained on the surface or inactivated due to these strong electrostatic forces.

The PTFE surface (Figure 5) displayed a sharp dose-response curve (R^2^ = 0.999) for UV doses 0–25 mJ/cm^2^. After a UV dose of 25 mJ/cm^2^, MS-2 inactivation down to the detection limit (5.8 log) was observed. These results differed from the other surfaces evaluated in that ceramic, Formica laminate and stainless steel did not reach inactivation to the detection limit, even at a UV dose of 100 mJ/cm^2^. In addition, the PTFE results varied from the large repository of UV dose-response curves for water inactivation tabulated by Malayeri et al. [13]. The water UV dose-response curve suggests that ~1.5 log reduction of MS-2 bacteriophage is observed at 25 mJ/cm^2^, compared to almost 6 log reduction observed on the PTFE surface [13]. 

This data may be explained by the high UV reflectance characteristic of PTFE surfaces. It has been reported that PTFE is over 97% reflective at UV_254_ [2,31]. The current knowledge suggests that reflectance is responsible for a higher level of inactivation, as it increases UV exposure to viral particles across the surface. The UV wavelengths have a higher probability of reaching the viral particle on the PTFE surface, as the UV light continuously reflects when meeting the surface. Therefore, the UV_254_ wavelength can move into the pore spaces and across the surface area more easily [2].

Additional work suggests that PTFE is highly reflective at wavelengths ranging from 250–500 nm. PTFE is thought to be a desirable material for UV devices due to its high reflectivity and resistance to degradation [32]. Another study compared the distribution of UV dose inside of a UV chamber using PTFE, acrylonitrile butadiene styrene *[sic]*, silver gloss self-adhesive aluminum, and Rosco matte black Cinefoil as UV reflectors. The study found that the PTFE reflector delivered the most even spread of irradiance to the test object inside the chamber [33]. Unlike vacuum UV at 185 nm UV wavelengths at 250 nm or greater do not significantly alter the chemical structure of the PTFE [34], therefore viral inactivation is attributed to the high reflectivity of UV_254_ identified in the literature [2,31,32,33].

In 2018, Mitchell et al. studied the effects of UV_254_ surface inactivation on Formica laminate and stainless steel [35]. The author tested three bacterial species and one virus, Murine Norovirus (MNV). The MNV achieved 2.85 log inactivation on the stainless steel surfaces, experiments on the Formica laminate were not conducted for MNV. After statistical modeling, it was determined that the stainless steel surface observed much higher inactivation levels than Formica laminate across all pathogens. It was proposed that this was due to the porous nature of the laminate surface. Stainless steel is a more polished surface, therefore likely did not allow for surrogate protection [35]. In our study, the surface porosity of Formica laminate and stainless steel used was found to be similar. The UV dose-response curves for MS-2 were similar for these two surfaces.

A 2022 study evaluated the efficacy of UV_254_ inactivation on glass, plastic, wood, stainless steel, and PPE products [36]. The authors used Bacteriophage Phi 6 as the surrogate at concentrations of approximately 10^8^ PFU/mL. Two UV devices were utilized with UV doses ranging from 0–300 mJ/cm^2^ and 0–240 mJ/cm^2^ for devices 1 (2.4 W) and 2 (5.5 W), respectively. The results of the stainless steel experiments were compared to MS-2 bacteriophage inactivation observed on stainless steel in our study by utilizing the polynomial regression equation displayed in Figure 6. The inactivation of MS-2 on stainless steel was higher than the inactivation of Phi 6 at comparative UV doses [30]. Masjoudi et al. [37] report that Phi 6 is more UV resistant than MS-2, therefore supporting the lower level of inactivation observed with Phi 6, when compared to MS-2 bacteriophage.

Additionally, our work reiterated the findings of this author and others mentioned above. Bartolomeu et al. [36] found that pore spaces shield viral particles from UV wavelengths, therefore, porous surfaces yield lower levels of viral inactivation. The author also reported that UV inactivation on the plastic surface yielded the lowest inactivation levels, followed by stainless steel and glass. The author did not explore why these variations were observed, although emphasized that the efficacy of UV surface inactivation varies by surface type and corresponding characteristics [36].

### Contact Angle, Surface Roughness, Surface Porosity and Zeta Potential

The recovery of MS-2 bacteriophage from the positive control experiments varied by surface type. Several mechanisms have been explored and provide insight into these variations. Characterization of anthropogenic, monolithic, surfaces is commonly performed both at the macroscale and microscale.

Understanding the interaction between viral particles and surfaces requires a more detailed understanding of microscale parameters including contact angle, SEM surface porosity and zeta potential [38,39]. Surface porosity is a measure of the porous (void) area divided by the surface area of the material. Zeta potential is a measure of the electrical potential at the slipping plane. The slipping plane is the boundary separating stationary liquid adhered to the surface, versus mobile fluid that moves freely [40]. The contact angle is a measure of hydrophobicity/hydrophilicity or wettability of a surface.

Our research demonstrates that MS-2 bacteriophage recovery is linearly related to the contact angle (Figure 7), as the contact angle increases, the molecular properties of the surface chemistry are more hydrophobic. This suggests that more hydrophobic surfaces have smaller areas for viral bonding, therefore allowing for easier viral recovery.

The zeta potential measurements, including the surface charge, were taken and are reported in Figure 11. This figure displays a linear, inverse relationship, such that as zeta potential increases, MS-2 recovery decreases. As zeta potential increases, the ionic bonds between the negatively charged MS-2 bacteriophage virus [41] and the surfaces strengthen, making viral recovery more challenging. Additionally, Appendix A, displays the relationship between contact angle and zeta potential. As the contact angle increases (hydrophobicity increases), zeta potential decreases. The combined forces of these surface characteristics synergistically impact the recovery of MS-2 bacteriophage.

These findings can be supported by the literature [42,43]. De Matteis et al. (2020) studied the contact angle of low-density polyethylene films and collected data on pristine polymers, polystyrene derivatives, sulfonated polyethylene, and carbonate sand. The results of this work confirmed our results, that the surface zeta potential decreases as the contact angle increases [42]. MS-2 is a negatively charged virus [41], therefore would have a stronger repulsion force to a negatively charged surface. Figure 7 displayed that more viral recovery occurred on the surfaces with the higher contact angles, caused by lesser viral sorption occurring on the surface. It follows that the negatively charged MS-2 [43] viral particles are repelled by the negative zeta potential surfaces and the viral particles are more easily recovered. The charge interaction also explains why the only surface studied with a positive zeta potential, aluminum, demonstrated complete viral retention.

Figure 9 and Figure 10 display the variation in surface porosity by surface type. Figure 10 displayed a strong correlation between viral recovery from the positive control and surface porosity, as surface porosity increased the recovery decreased. Jaffe [44] explored the effects of higher surface porosity on UV_254_ disinfection and has named the phenomenon the “canyon wall effect”. Surfaces with larger pore sizes offer protection to viral particles residing in those pores, as the approximate diameter of an MS-2 viral particle is 27 nm [45], which is significantly smaller than the size of the pores within the surface [44]. The pores on the surfaces allow the viral particles to be shadowed by the UV wavelengths. Therefore, a portion of the virus applied to the surface is protected, and when recovered remains infective. These results were observed with the ceramic surface. As shown in Figure 3, this surface approached its maximum level of inactivation at approximately 75 mJ/cm^2^, due to the ceramic’s large pore sizes.

This work demonstrates that the MS-2 recovery is inversely, linearly, proportional to the SEM surface porosity (Figure 10). These results suggest that materials with higher surface porosity have more surface area available for viral particle adsorption and may further explain the results observed with the highly oxidized aluminum surface. The aluminum surface demonstrated the highest surface porosity of all materials, therefore allowing more surface area for the virus to adsorb.

Similarly, this research demonstrates that the higher the contact angle (hydrophobicity) of the material, the lower the surface porosity (Appendix A), resulting in synergy between two factors both promoting increased viral recovery. This correlation also suggests that materials with higher surface porosity values are more absorbent, thus reducing the contact angle.

Appendix A displays the absolute zeta potential versus the surface porosity of each material. This figure displays a non-linear relationship, such that as surface porosity increases, zeta potential decreases. This phenomenon is supported by Yakin et al. [46] and can be explained by the formation of an electrical double-layer overlap. Smooth surfaces or surfaces with small pores yield an overlap of electrical potentials similar to the zeta potential, whereas large pore spaces lead to divergence of the electrical potentials [46]. The data in Appendix A supports this theory well. As shown in Figure 10, for materials with low surface porosity (PTFE, stainless steel, and Formica laminate) viral recovery is governed by the other surface characteristics (contact angle, zeta potential) discussed above.

The macroscale measurement of surface roughness (Rs) determines the irregularities of peaks and troughs along a surface [47]. This measurement provides general information for comparisons and preliminary characterization but is of limited value for understanding the surface behavior of relatively smooth, anthropogenic, surfaces [38]. Measurements made in this research (Appendix A) confirm that there is no statistical correlation between surface roughness and SEM surface porosity for the materials tested since they are all in the very smooth to smooth range (Rs 0.004 to 1.675 μm range).

Figure 8 displayed a strong correlation between surface roughness and concentration (PFU/mL) of MS-2 recovered from the positive controls. When comparing Figure 7 and Figure 8, an inverse relationship between surface roughness and contact angle exists. As surface roughness increases, the contact angle decreases. This relationship has also been observed by other researchers, specifically for hydrophilic surfaces [48]. From the contact angles, it can be determined that aluminum, ceramic, Formica laminate, and stainless steel are hydrophilic surfaces. Therefore, the findings of Li (2021) relate to the surfaces used for these experiments. The inverse relationship between surface roughness and contact angle may be responsible for the strong correlation also observed between surface roughness and viral recovery.

A report from the Massachusetts Institute of Technology (MIT) [49] relating surface roughness to contact angle indicates that surface roughness increases the hydrophilic/hydrophobic properties of the surface. Increasing the roughness of hydrophobic surfaces increases the hydrophobic interactions occurring on the surface. While increasing the roughness of a hydrophilic surface decreases its hydrophilicity. A recent study by Du et al. [50] confirms this phenomenon but states that the degree of roughness and hydrophilicity of the surface can make these results highly variable.

These findings are reinforced by Appendix A. A correlation between surface roughness and contact angle was not observed for aluminum and stainless steel. Although, a linear relationship for PTFE, Formica laminate and ceramic was observed. Therefore, it follows that as the surface roughness increased, they linearly became more hydrophilic. Thus, the recovery of MS-2 was lowest for ceramic followed by Formica laminate and then PTFE which had low surface roughness, strong hydrophobic surface behavior and the most MS-2 recovery.

## 5. Conclusions

This research revealed several major findings regarding the efficacy of UV_254_ for surface inactivation, which are listed below.

Experimental results of the highly oxidized aluminum surfaces demonstrated that interactions between the MS-2 bacteriophage virus and the aluminum surface caused retention/inactivation down to the detection limit, independent of UV dose.The UV dose (25 mJ/cm^2^) required to inactivate MS-2 to non-detectable levels on the PTFE surface was significantly lower than the ceramic, Formica laminate, and stainless steel surfaces (Appendix A).A strong (R^2^ = 0.955) linear correlation between contact angle and MS-2 recovery from the surfaces was discovered. These data demonstrate that as the contact angle increased, the recovery of MS-2 from the surfaces increased.An inverse relationship between contact angle and surface roughness was observed. A strong linear correlation (R^2^ = 0.999) between PTFE, Formica laminate, and ceramic was discovered. As surface roughness increased, viral recovery decreased. Whereas, as the contact angle increased, viral recovery increased.When analyzed with SEM, it became apparent that the surface porosity varied between each material evaluated. The aluminum and ceramic had the surface highest porosity values, 25%, and 17.3%, respectively. Formica laminate, PTFE, and stainless steel all had significantly lower surface porosities, 2.3%, 1.6%, and 2.9%, respectively. A strong correlation was identified between surface porosity and viral retention from the materials, such that as surface porosity increased, viral recovery decreased.A strong, inverse, linear correlation was observed between zeta potential and MS-2 recovery (R^2^ = 0.903). As zeta potential increased, MS-2 bacteriophage recovery decreased.As determined by equivalency testing, the UV dose-response curves were not equivalent for any of the surfaces evaluated. The UV dose response varied as a function of surface type.

## Figures and Tables

**Figure 1 microorganisms-11-02157-f001:**
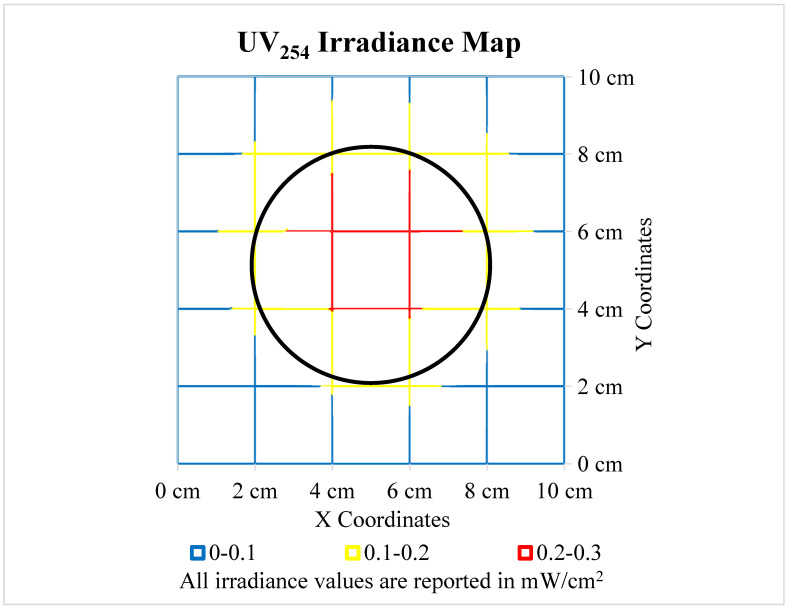
Figure 1 displays the spread of UV_254_ irradiance by the Trojan UV collimated beam. An IL1700 radiometer was used to collect these data. The collimated beam was 9 cm above the radiometer for these measurements. The weighted average irradiance at this height was 0.203 mW/cm^2^ inside of the black circle, which represents the outline of the surfaces.

**Figure 2 microorganisms-11-02157-f002:**
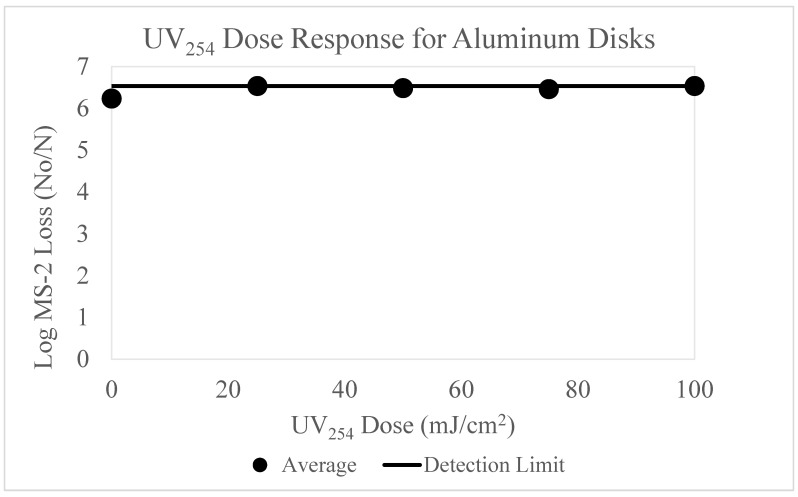
Figure 2 displays the UV_254_ dose-response curve for aluminum disks. The *y*-axis displays the log MS-2 bacteriophage loss and the *x*-axis displays the UV_254_ doses used for experimentation (0, 25, 50, 75 and 100 mJ/cm^2^). The inoculum concentration applied to each surface was ~10^9^ PFU/mL. Three trials testing the five UV doses were conducted. The detection limit is displayed as a line on the graph.

**Figure 3 microorganisms-11-02157-f003:**
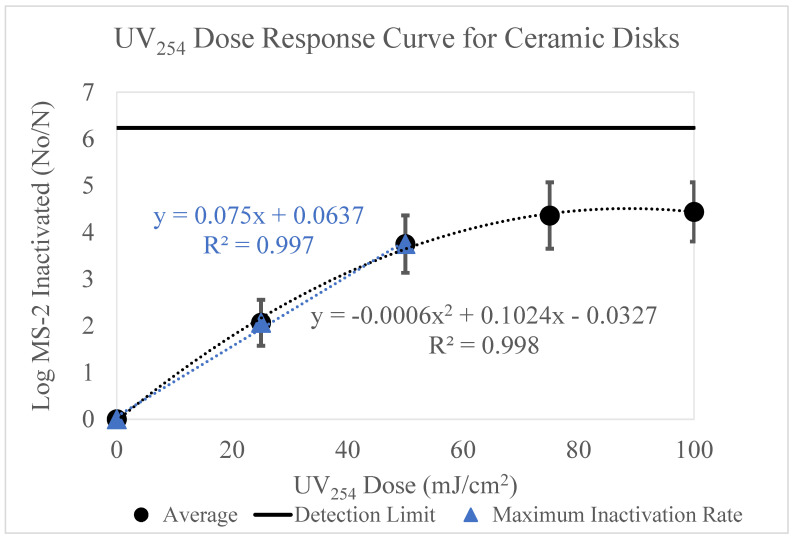
Figure 3 displays the UV_254_ dose-response curve for ceramic disks. The *y*-axis displays the log MS-2 bacteriophage loss and the *x*-axis displays the UV_254_ doses used for experimentation (0, 25, 50, 75 and 100 mJ/cm^2^). The inoculum concentration applied to each surface was ~10^9^ PFU/mL. Three trials testing the five UV doses were conducted. The detection limit is displayed as a line on the graph. The maximum inactivation rate is shown by the blue linear trendline.

**Figure 4 microorganisms-11-02157-f004:**
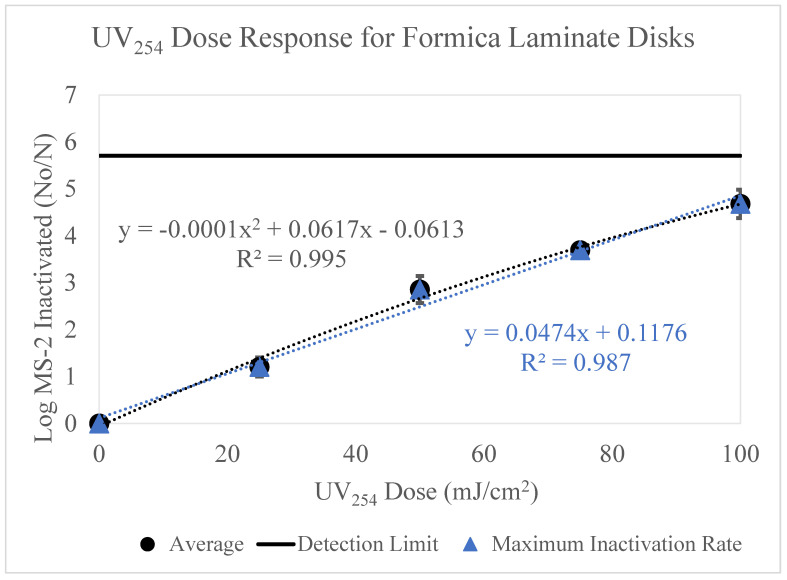
Figure 4 displays the UV_254_ dose-response curve for Formica Laminate disks. The *y*-axis displays the log MS-2 bacteriophage loss and the *x*-axis displays the UV_254_ doses used for experimentation (0, 25, 50, 75 and 100 mJ/cm^2^). The inoculum concentration applied to each surface was ~10^9^ PFU/mL. Three trials testing the five UV doses were conducted. The detection limit is displayed as a line on the graph. The maximum inactivation rate is shown by the blue linear trendline.

**Figure 5 microorganisms-11-02157-f005:**
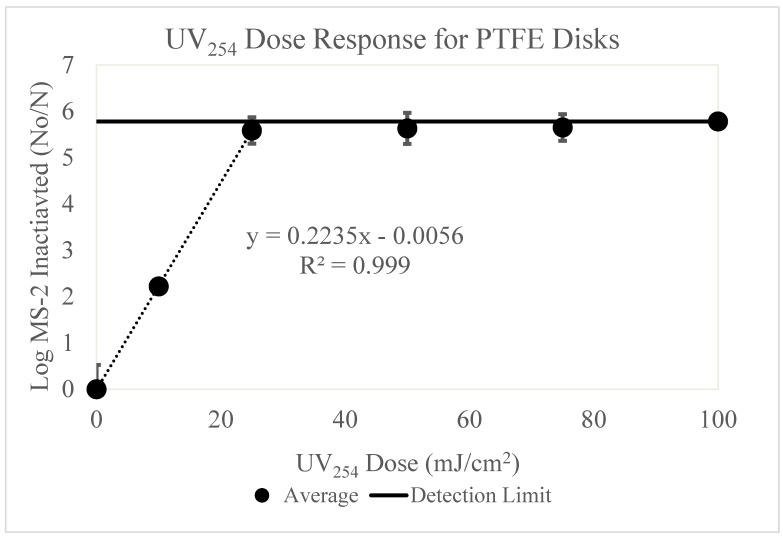
Figure 5 displays the UV_254_ dose-response curve for PTFE disks. The *y*-axis displays the log MS-2 bacteriophage loss and the *x*-axis displays the UV_254_ doses used for experimentation (0, 25, 50, 75 and 100 mJ/cm^2^). The inoculum concentration applied to each surface was ~10^9^ PFU/mL. Three trials testing the five UV doses were conducted. Additional trials at 10 mJ/cm^2^ were conducted to complete the dose-response curve. The detection limit is displayed as a line on the graph.

**Figure 6 microorganisms-11-02157-f006:**
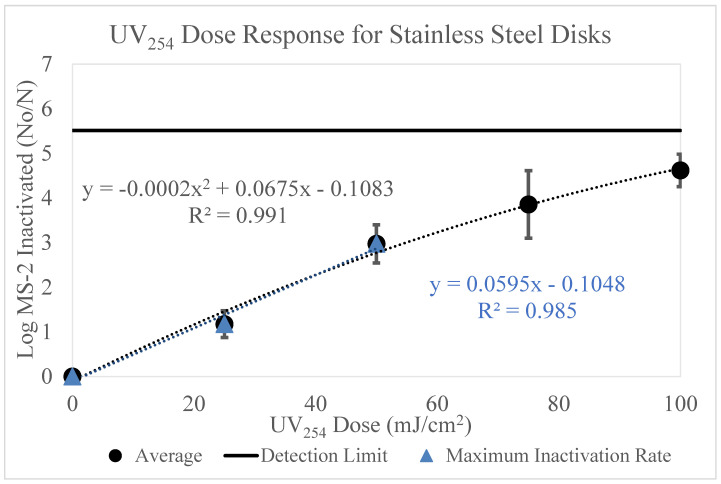
Figure 6 displays the UV_254_ dose-response curve for stainless steel disks. The *y*-axis displays the log MS-2 bacteriophage loss and the *x*-axis displays the UV_254_ doses used for experimentation (0, 25, 50, 75 and 100 mJ/cm^2^). The inoculum concentration applied to each surface was ~10^9^ PFU/mL. Three trials testing the five UV doses were conducted. The detection limit is displayed as a line on the graph. The maximum inactivation rate is shown by the blue linear trendline.

**Figure 7 microorganisms-11-02157-f007:**
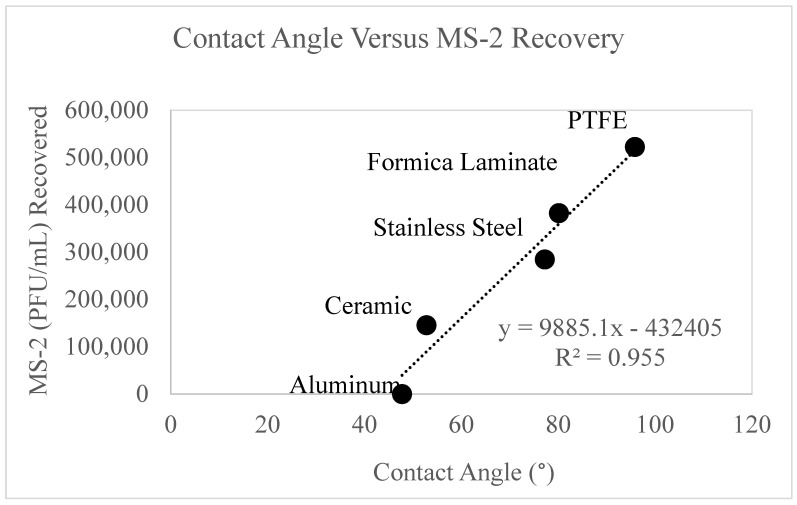
displays MS-2 bacteriophage recovery as a function of contact angle. The *y*-axis displays the concentration of MS-2 bacteriophage recovered and the *x*-axis displays the contact angle. The contact angle for each surface is reported on the figure and labeled accordingly.

**Figure 8 microorganisms-11-02157-f008:**
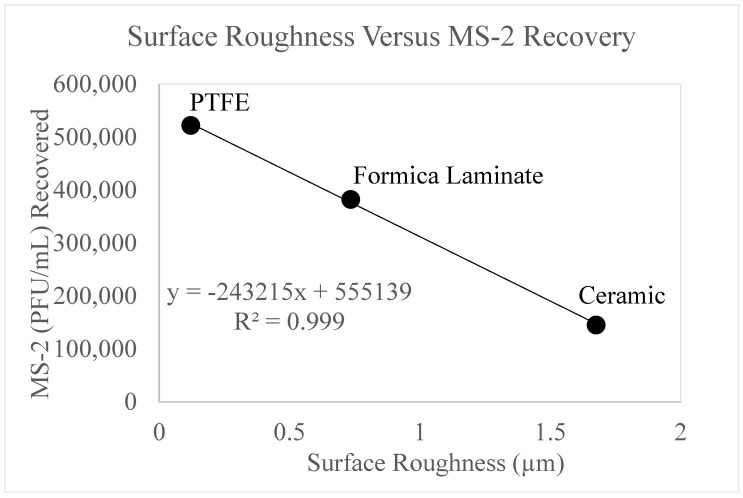
displays the MS-2 bacteriophage recovery as a function of surface roughness. The *y*-axis displays the concentration of MS-2 bacteriophage recovered and the *x*-axis displays the surface roughness in µm. The surface roughness for PTFE, stainless steel, and ceramic are reported on the figure and labeled accordingly.

**Figure 9 microorganisms-11-02157-f009:**
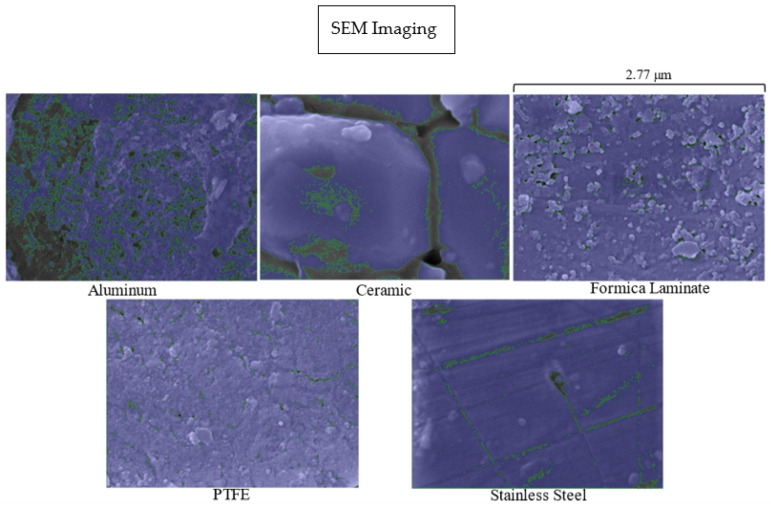
displays images taken using SEM. The images were taken at 100,000 times magnification for each surface. The dark spots highlighted in green represent pore spaces.

**Figure 10 microorganisms-11-02157-f010:**
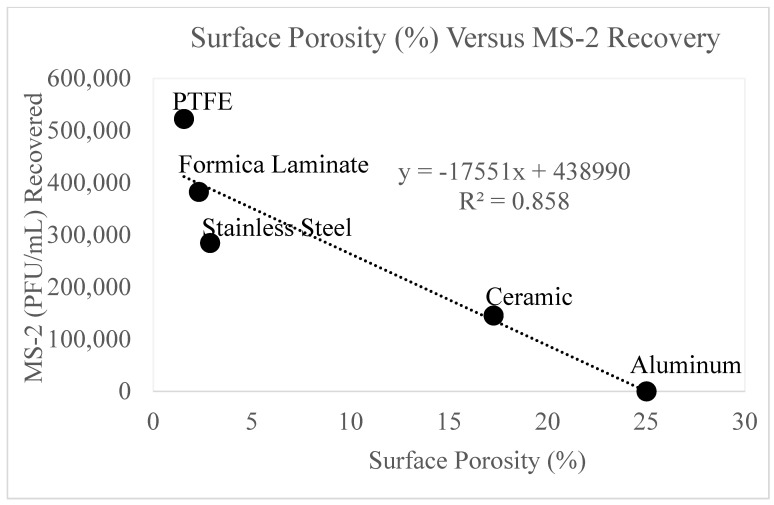
displays the MS-2 bacteriophage recovery as a function of surface porosity. On the *y*-axis is MS-2 bacteriophage recovery from the positive control (0 mJ/cm^2^). On the *x*-axis is surface porosity as a percentage. The surfaces are labeled accordingly in the figure.

**Figure 11 microorganisms-11-02157-f011:**
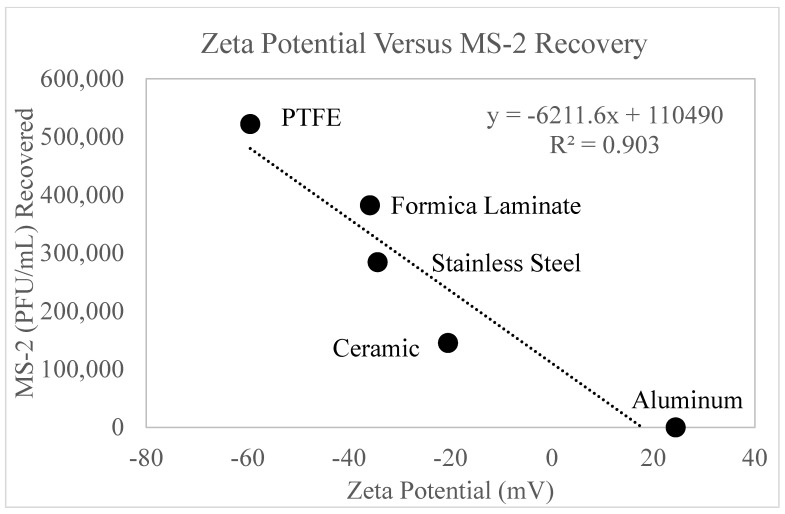
displays the MS-2 bacteriophage recovery as a function of zeta potential. On the *y*-axis is MS-2 bacteriophage recovery from the positive control (0 mJ/cm^2^). On the *x*-axis is zeta potential (mV). The surfaces are labeled accordingly in the figure.

**Table 1 microorganisms-11-02157-t001:** Surfaces Utilized for Experimentation.

Surface Type	Manufacturer/Description
Aluminum	6061 Aluminum Sheet (Lostronaut, Vancouver, WA, USA)—Finely Polished and Deburred Prior to Initial Use
Ceramic	Ceramic Solutions (Conroe, TX, USA): 99.8% Alumina Discs
Formica Laminate	Lowes (Mooresville, NC, USA): 6321-43 Oxidized Maple
PTFE	Thorlabs (Newton, NJ, USA): Fabricated from sintered, crystalline, fused, and skived, virgin PTFE.
Stainless Steel	MetalsDepot (Winchester, KY, USA): 304 Stainless with Mirror Finish (#8)

Table 1 displays the specifications and manufacturer information about each of the surfaces used.

**Table 2 microorganisms-11-02157-t002:** UV Doses and Exposure Times.

Doses (mJ/cm^2^)	0	25	50	75	100
Exposure Time (s)	0	123.2	246.3	369.5	492.6
Exposure Time (min)	0	2.05	4.11	6.16	8.21

Table 2 displays the UV doses and corresponding exposure times utilized for experimentation.

**Table 3 microorganisms-11-02157-t003:** Tabulated data for each surface.

Surface	k Value (cm^2^/ mJ) Polynomial	k Value (cm^2^/mJ) MIR	Contact Angle (°)	Surface Porosity (%)	Surface Roughness (µm)	Zeta Potential (mV)
Aluminum	NM	NM	47.8	25.02	0.24	24.4
Ceramic	0.0424	0.075	52.8	17.26	1.675	−20.5
Formica Laminate	0.0517	0.0474	80.1	2.31	0.734	−35.9
PTFE	0.2235	0.2235	95.8	1.56	0.121	−59.5
Stainless Steel	0.0475	0.0595	77.2	2.88	0.004	−34.4

Table 3 displays the average k values of the UV dose response curves, contact angle, surface porosity, surface roughness and zeta potential for each surface. NM indicates that the value was not measured. MIR stands for Maximum Inactivation Rate.

## Data Availability

For a complete report of all raw data collected during experimentation please visit: doi:10.34051/d/2022.4 (accessed on 1 March 2023).

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
