# Peer review of "Virus Behavior after UV254 Treatment of Materials with Different Surface Properties"

_microorganisms, 2023, doi:10.3390/microorganisms11092157_

Round 1

Reviewer 1 Report

Very nice study with some interesting outcomes. I have a few suggestions for improving the manuscript.

define abbreviations at first use. For example, IUVA should be defined at first use at line 36 and consider defining the other acronyms there.

Line 37: Credit for the work should be given to the authors, not the journal (Poster et al. instead of Journal of NIST). Similarly, on line 99 give credit to Kowalski, not "the literature", and on line 593 give credit to Masjoudi et al, not "the literature". Please take a look throughout the manuscript and ensure that you give credit to authors.

line 63: I think you mean standardized, not standard

lines 65 through 91: Please cite the reference at the beginning of the paragraphs so readers know where the information is coming from. The citations at the end of the paragraphs can be retained as well.

line 97: Please reword to make this a sentence.

line 105: consider "research addresses" instead of "research studies"

line 203: Not sure what is meant here. Is the UV used to kill any remaining live E. coli? If so this should be stated, along with the fluence applied.

Lines 213 -218: Inactivation of viruses is typically first-order with respect to fluence. A polynomial line fits the data but makes no scientific sense. For instance, take your equation for ceramic and plug in a fluence of 200 mJ/cm2. You will see that the polynomial equation will return a negative result, which is impossible. I suggest evaluating the reasons behind the leveling off (perhaps running out of virus) and then trying to establish a linear portion of the curves as you have done for figure 5 for calculating rate constants (log10 reduction per mJ/cm2).

Line 597: I would say that your data reiterate the findings of Bartolomeu, not vice versa.

As a general comment, the discussion is very long. Please try to shorten it by not repeating results unless needed to set discussion text. I know this is difficult, but try to focus the reader's attention to the main points you wish to make.

I have identified some suggestions above

Author Response

Below are the author’s responses to reviewer 1  comments and suggestions. We greatly appreciate the thorough review and this opportunity to improve this manuscript. We have addressed all comments to the best of our ability.

Reviewer

Comment

Author Response

1

Define abbreviations at first use. For example, IUVA should be defined at first use at line 36 and consider defining the other acronyms there.

The full name of each organization has been added to the text.

1

Line 37: Credit for the work should be given to the authors, not the journal (Poster et al. instead of Journal of NIST). Similarly, on line 99 give credit to Kowalski, not "the literature", and on line 593 give credit to Masjoudi et al, not "the literature". Please take a look throughout the manuscript and ensure that you give credit to authors.        

These citations have been updated and the rest of the text has been checked to ensure this issue was corrected in any other locations. In one case (citation 49), MIT is mentioned as the author. This source only provided MIT for author information, as it is a web link.

1

line 63: I think you mean standardized, not standard.

The text has been updated to use the term standardize instead of standard.

1

lines 65 through 91: Please cite the reference at the beginning of the paragraphs so readers know where the information is coming from. The citations at the end of the paragraphs can be retained as well.

The citations have been added at the beginning of the paragraphs. Additional corrections were made for other places where this error occurred in the manuscript.

1

line 97: Please reword to make this a sentence.

This sentence has been revised and now reads as a full sentence.

1

line 105: consider "research addresses" instead of "research studies"

"Research studies" has been changed to "research addresses"

1

line 203: Not sure what is meant here. Is the UV used to kill any remaining live E. coli? If so, this should be stated, along with the fluence applied.

The text was edited to make this clearer.

1

Lines 213 -218: Inactivation of viruses is typically first-order with respect to fluence. A polynomial line fits the data but makes no scientific sense. For instance, take your equation for ceramic and plug in a fluence of 200 mJ/cm2. You will see that the polynomial equation will return a negative result, which is impossible. I suggest evaluating the reasons behind the leveling off (perhaps running out of virus) and then trying to establish a linear portion of the curves as you have done for figure 5 for calculating rate constants (log10 reduction per mJ/cm2).

Thank you for this comment. Several changes have been made to the manuscript to address these concerns. A linear portion of all appropriate graphs (ceramic, Formica laminate, and stainless steel) has been created and the 'maximum inactivation rate' is now reported in Table 3.  The polynomial trendlines were kept for these figures as they display the effects of tailing which relate directly to the effects of surface properties, and it is important for those to be quantified so rates calculated from the polynomial trendlines are used. The authors appreciate the reviewers’ concern about running out of virus, but the detection limit was labeled on all of the dose-response curves. These figures displayed that running out of virus did not occur since the leveling off is occurring well before this detection limit is reached.

1

Line 597: I would say that your data reiterate the findings of Bartolomeu, not vice versa.

The wording of this sentence has been changed per the reviewer’s suggestion.

1

As a general comment, the discussion is very long. Please try to shorten it by not repeating results unless needed to set discussion text. I know this is difficult but try to focus the reader's attention to the main points you wish to make.

This comment was addressed by the journal’s academic editor.

Reviewer 2 Report

The authors describe a relevant study that contributes to a better understanding of the challenges of UV disinfection of surfaces with an unexpected result for aluminum, even without UV. The manuscript is well written and easy to follow.

There are couple of places where the manuscript can be improved based on the following comments:

Definitions:

·       The authors are linking “porosity” in this study to the surface porosity (and not the material porosity that is more commonly used). It would be helpful to change the header in Table 3 from “Porosity” to “Surface Porosity” as well as in Figure 10 [title and x-axis title] to avoid confusion with the volume-based porosity definition of a material. There are other places in the manuscript (e.g., lines 452/453) where the text connects the measured porosity to the material instead of the surface of the material.

·       Please verify if the definition of porosity from reference 34 is correct. What is meant by the “total volume of a surface”? A surface cannot be defined by a volume unless the third dimension (thickness of a material) is included. Instead, the surface porosity definition as used in the calculation should have been noted/cited as surface ratio.

Test methods:

·       It would be helpful to the reader to know if there was a delay in time between the application via a swab and the t=0 min start of the UV exposure. Did the inoculum dry or was it still wet at the start? According to the recent literature, there is a significant difference between dose responses for wet inoculum and dried inoculum, including for MS-2. Dried inoculum may have a closer resemblance to a real contaminated surface that need to be disinfected; such reflection and associated references are missing from the manuscript.

·       The contact angle measurement must have been done by placing a droplet of the MS-2 inoculum on the surface (no size/volume information is provided). While this is a valid method, the inoculum application using a swab and a stencil may not result in the same type of droplets. Is there visual evidence that the swab application led to similar droplet characteristics?

·       The contact angle would be linked to the properties of the inoculum. This should be noted as a limitation of the study.   

·       The calculated log reductions appear to have been calculated against the recovery of MS-2 at t=0. It would have been more informative to consider inclusion of a non-UV exposure positive control test with recoveries at all pre-determined time points. This would eliminate the inclusion of natural decay in the presented data. Can the authors provide insight on the % loss of MS-2 over time?

Results:

·       The results for aluminum are quite remarkable. Did the authors consider swabbing and rinsing of the aluminum surface without MS-2 on it? A spike of MS-2 added to that extraction volume followed by the same analysis / enumeration steps would distinguish between full inactivation or the inability to remove MS-2 from the surface.

·       As noted by the authors, the aluminum results are attributed to the oxidation at the surface due to repeated disinfection cycles (lines 501-503) as per current CoVID-19 cleaning protocols. The authors also write (line 180-182) that an autoclave was used to sterilize some of the materials, including aluminum. Did the autoclave cause a similar level of oxidation? It would be interesting to investigate this 6061 aluminum in more detail. The manuscript refers to “finely polished” in Table 1. Was that the description of the material when purchased / as received or was this polishing done to the material prior to use. If so, it should be described in more detail.

Manuscript edits

·       Line 36, IUVA should be spelled out at first use; not in line 49. Similarly, ASHRAE, ANSI and NIST are not common abbreviations for a significant portion of the microorganisms readership.

References:

·       Reference 16 should be removed and replaced with a in text reference to the vendor of the software; the interview reference has no relevance to the reader. If the selection of the software package came from S. Verhoeven, he can be recognized in the acknowledgement section.

·       Reference 17 has no value to the reader unless P Ramsey was the one who discovered the described approach. If that is the case, it should be linked to a publicly available publication by P Ramsey

Author Response

Below are the author’s responses to reviewer 2 comments and suggestions. We greatly appreciate the thorough review and this opportunity to improve this manuscript. We have addressed all comments to the best of our ability.

Reviewer

Comment

Author Response

2

Definitions: The authors are linking “porosity” in this study to the surface porosity (and not the material porosity that is more commonly used). It would be helpful to change the header in Table 3 from “Porosity” to “Surface Porosity” as well as in Figure 10 [title and x-axis title] to avoid confusion with the volume-based porosity definition of a material. There are other places in the manuscript (e.g., lines 452/453) where the text connects the measured porosity to the material instead of the surface of the material.

Thank you for this comment. The manuscript has been revised to provide clarity that the author is referring to surface porosity, rather than material porosity.

2

Definitions: Please verify if the definition of porosity from reference 34 is correct. What is meant by the “total volume of a surface”? A surface cannot be defined by a volume unless the third dimension (thickness of a material) is included. Instead, the surface porosity definition as used in the calculation should have been noted/cited as surface ratio.

The definition of surface porosity has been updated in the text. Reference 34 was removed as the description from the methods section sufficiently defined surface porosity.

2

Test Methods: It would be helpful to the reader to know if there was a delay in time between the application via a swab and the t=0 min start of the UV exposure. Did the inoculum dry or was it still wet at the start? According to recent literature, there is a significant difference between dose responses for wet inoculum and dried inoculum, including for MS-2. Dried inoculum may have a closer resemblance to a real contaminated surface that need to be disinfected; such reflection and associated references are missing from the manuscript.

Thank you for this comment. The methods section has been updated to explain that the surfaces were not dried prior to treatment. We also tied in current literature to explain findings of inoculum drying and that the results may have been different if the surfaces were dried prior to UV exposure.

2

 Test Methods: The contact angle measurement must have been done by placing a droplet of the MS-2 inoculum on the surface (no size/volume information is provided). While this is a valid method, the inoculum application using a swab and a stencil may not result in the same type of droplets. Is there visual evidence that the swab application led to similar droplet characteristics?

This comment was partially addressed by the journal’s academic editor. The swabbing method formed droplets on the surface of the materials. The droplet volume used to measure the contact angle was added to the materials section.

2

Test Methods: The contact angle would be linked to the properties of the inoculum. This should be noted as a limitation of the study.

This comment was addressed by the journal’s academic editor.

2

Test Methods:  The calculated log reductions appear to have been calculated against the recovery of MS-2 at t=0. It would have been more informative to consider the inclusion of a non-UV exposure positive control test with recoveries at all pre-determined time points. This would eliminate the inclusion of natural decay in the presented data. Can the authors provide insight on the % loss of MS-2 over time?

Thank you for pointing out this important issue. The exposure times used for these experiments were short, the highest UV dose correlated to an ~ 8 min exposure time.  The author has worked with other wavelengths (405 nm), which require significantly longer dosing times on the order of hours where this issue can be quite significant. The author has observed viral decay for these experiments, starting at exposure times of multiple hours. Additionally, due to the high inactivation achieved on the PTFE surface, the author ran experiments to determine if viral decay on this surface was a contributing factor to viral loss. This experiment was conducted for the longest exposure time (~8 mins), and viral decay was not observed. These data can be seen in the compiled raw data DOI link listed in the manuscript. Per UNH policy, final updates to the DOI link will be made when the paper is accepted for publication.

2

Results: The results for aluminum are quite remarkable. Did the authors consider swabbing and rinsing of the aluminum surface without MS-2 on it? A spike of MS-2 added to that extraction volume followed by the same analysis / enumeration steps would distinguish between full inactivation or the inability to remove MS-2 from the surface.

Thank you for this comment and the suggestion about this very interesting experiment. Unfortunately, this experiment was not conducted. The funding for this project is complete, therefore we cannot perform further experiments at this time. This suggestion will be recommended for future work of our laboratory group.

2

Results:  As noted by the authors, the aluminum results are attributed to the oxidation at the surface due to repeated disinfection cycles (lines 501-503) as per current CoVID-19 cleaning protocols. The authors also write (line 180-182) that an autoclave was used to sterilize some of the materials, including aluminum. Did the autoclave cause a similar level of oxidation? It would be interesting to investigate this 6061 aluminum in more detail. The manuscript refers to “finely polished” in Table 1. Was that the description of the material when purchased / as received or was this polishing done to the material prior to use. If so, it should be described in more detail.

The manuscript was edited to clean up the poor wording. The oxidation of the aluminum surface is thought to be from autoclaving. Table 1 has been updated to indicate that the aluminum was finely polished prior to use.

2

Manuscript Edits:  Line 36, IUVA should be spelled out at first use; not in line 49. Similarly, ASHRAE, ANSI and NIST are not common abbreviations for a significant portion of the Microorganism’s readership.

The text has been revised such that acronyms are spelled out by the first mention in the text.

2

References:  Reference 16 should be removed and replaced with a in text reference to the vendor of the software; the interview reference has no relevance to the reader. If the selection of the software package came from S. Verhoeven, he can be recognized in the acknowledgement section.

Reference 16 was deleted, the EPA guidance manuals that S. Verhoeven obtained this information from were cited instead.

2

References: Reference 17 has no value to the reader unless P Ramsey was the one who discovered the described approach. If that is the case, it should be linked to a publicly available publication by P Ramsey.

This citation was removed. P Ramsey did not discover the described approach. A more appropriate citation was added.

Round 2

Reviewer 1 Report

The revisions are acceptable. Please check figure legends lines 326, 347, 391 for duplication of "by the"